# Multivalent Pyrrolidine Iminosugars: Synthesis and Biological Relevance

**DOI:** 10.3390/molecules27175420

**Published:** 2022-08-24

**Authors:** Yali Wang, Jian Xiao, Aiguo Meng, Chunyan Liu

**Affiliations:** 1College of Pharmacy, North China University of Science and Technology, Tangshan 063000, China; 2Affiliated Hospital, North China University of Science and Technology, Tangshan 063000, China

**Keywords:** iminosugar, pyrrolidine, multivalent effect, glucosidase inhibitors

## Abstract

Recently, the strategy of multivalency has been widely employed to design glycosidase inhibitors, as glycomimetic clusters often induce marked enzyme inhibition relative to monovalent analogs. Polyhydroxylated pyrrolidines, one of the most studied classes of iminosugars, are an attractive moiety due to their potent and specific inhibition of glycosidases and glycosyltransferases, which are associated with many crucial biological processes. The development of multivalent pyrrolidine derivatives as glycosidase inhibitors has resulted in several promising compounds that stand out. Herein, we comprehensively summarized the different synthetic approaches to the preparation of multivalent pyrrolidine clusters, from total synthesis of divalent iminosugars to complex architectures bearing twelve pyrrolidine motifs. Enzyme inhibitory properties and multivalent effects of these synthesized iminosugars were further discussed, especially for some less studied therapeutically relevant enzymes. We envision that this comprehensive review will help extend the applications of multivalent pyrrolidine iminosugars in future studies.

## 1. Introduction

Iminosugars, containing an endocycling nitrogen atom that effectively mimics carbohydrates by facilitating the reversible and competitive inhibition of their processing enzymes, have generated much attention in recent years as targets in the treatment of a wide range of illnesses (e.g., diabetes, cancer, tuberculosis, and lysosomal storage disorders, etc.) [1,2,3,4,5,6]. Given the enormous range of biochemical events in which carbohydrate processing enzymes are implicated, iminosugars have enormous potential to be developed as inhibitors of glycosidases (glycoside hydrolases), glycosyltransferases (glycoside synthases), metalloproteinases, and nucleoside-processing enzymes [7,8,9,10,11]. It is worth noting that structural modifications to find potent inhibitors of the above enzymes among two of the most studied classes of iminosugars, polyhydroxylated pyrrolidines and piperidines, arouse great interest because of the wide range of their biological properties, such as glycosidase inhibition, shown over the past five decades [12,13,14,15,16].

The most famous representative iminosugars belonging to piperidine are derivatives of 1-deoxynojirimycin (DNJ, Figure 1). Since DNJ’s isolation from white mulberry root bark in 1976, hundreds of artificial iminosugars based on DNJ have been synthesized and their bioactivities evaluated [17,18,19,20,21]. Two *N*-alkylated DNJ derivatives are approved drugs, *N*-hydroxyethyl-1-deoxynojirimycin (Miglitol, Figure 1) to treat type II diabetes, and *N*-butyl-1-deoxynojirimycin (Miglustat, Figure 1) to treat lysosomal storage disorders (e.g., Gaucher disease). Similarly, five-membered iminocyclitols, also known as pyrrolidine iminosugars, exhibited excellent inhibition toward glycosidases. For example, 2,5-dihydroxymethyl-3,4-dihydroxypyrrolidine (DMDP, Figure 1), the first pyrrolidine iminosugar extracted from the leaves of *Derris elliptica* in 1976 [22], proved to be a potent glycosidase inhibitor; subsequently, its analogs were also found to have significant effects on glycosidases [23,24,25]. 1,4-Dideoxy-1,4-imino-d-arabinitol (DAB, Figure 1), isolated from the fruit of *Angylocalyx boutiqueanus*, exhibited strong inhibition of glycogen phosphorylase, and is currently being explored for the treatment of type II diabetes [26,27]. 1,4-Dideoxy-1,4-imino-l-arabinitol (LAB, Figure 1), the enantiomer of DAB, displayed more potent specific glycosidase inhibition. A new α-glucosidase inhibitor based on LAB was reported by Kato et al. in 2012, which showed huge potential in reducing elevated plasma glucose after food intake when tested in vivo with a carbohydrate load at doses approximately ten times lower than the required dose of miglitol [27,28,29,30]. In addition, Radicamines A and B have attracted extensive interest because of their potent inhibition of α-glucosidases and potential pharmaceutical applications [31,32,33]. Overall, ample evidence has established that iminosugars have anti-diabetic effects [34,35,36,37,38]. Both classes of iminosugar derivatives would be promising drug candidates, and therefore the development of synthetic strategies and the evaluation of bioactivities are of decisive importance.

However, as previously introduced, only a few drugs are on the market. New strategies for developing iminosugar-based glycosidase inhibitors to understand vital biological processes or as clinical candidates are therefore major challenges in both academia and the pharmaceutical industry [39,40,41]. In the last decade, a multivalent glycosidase inhibition effect, which has been extensively used in developing lectin inhibitors to seek new therapeutic opportunities for carbohydrate-related diseases, was found and rapidly developed [42,43,44,45,46,47,48,49]. Johns and Johnson first reported the synthesis of divalent iminosugars and explored the contribution of the multivalent strategy to biological activity in 1998 [50]. Probably due to the reported multivalent compounds, which could not exhibit the expected inhibition ability to glycosidase enzymes, the design of glycosidase inhibitors has not been able to attract the interest of researchers [51,52]. The factors that hampered the application of multivalency to iminosugars may be as follows: Firstly, there is an intrinsic structural difference between glycosidases and lectins. The surface of lectins shows multiple carbohydrate-binding sites, while glycosides and other carbohydrate-processing enzymes are usually monomeric and therefore bind relatively weakly to multivalent substrates [53,54,55]. Secondly, the synthesis of multivalent iminosugars is challenging. Since the click reaction was immature before 2001, it was particularly hard to graft several monomers to a skeleton simultaneously [56,57]. In addition, the experimental results obtained were not encouraging [50,51,52]. It was not a rapidly emerging area with exciting potential until the discovery of a small but quantitative multivalent effect in α-mannosidase inhibition [58]. Based on the extensive literature involving lectins and glycoclusters, several potential interactions have been proposed to explain the multivalent effect. The “bind-and-recapture” process is a classical mode due to the increased concentration of active molecules concentration in proximity to the binding site (Figure 2a). The chelate effect can occur when the enzyme presents more than one active site (Figure 2b). In addition, stronger interactions will occur when some non-catalytic subsites interact with glycoclusters (Figure 2c). Moreover, cross-linking and aggregation processes may prevail with glycoclusters if the enzyme possesses a multimetric nature (Figure 2d) [59,60].

The construction of multivalent iminosugars follows conventional strategies, including modifications and protecting group chemistry of the iminosugars [50,61,62], coupling reactions using click chemistry [45,48,63], and recently developed supramolecular self-assembly based on π–π stacking or hydrophobic interactions [64,65,66]. Multimerization of the pyrrolidines using these strategies resulted in some interesting results. For example, multimeric pyrrolidine iminosugars were reported to be the first example of multivalent enhancers of human α-galactosidase A (α-Gal A), an enzyme involved in Fabry disease [49]. Multivalent dendrimers decorated with the DAB exhibited a relevant multivalent effect toward the lysosomal enzyme *N*-acetylgalactosamine-6-sulfatase (GALNS), which is involved in a rare metabolic disorder [67]. Other multivalent effects of multivalent pyrrolidine iminosugars were obtained with carbohydrate-active enzymes such as Golgi α-mannosidase II [68], β-*N*-acetylglucosaminidase [69], and α-l-fucosidase [70]. This unique class of compounds may provide new pharmaceutical opportunities to treat diseases involving carbohydrate-processing enzymes.

Several reviews on the topic of multivalent iminosugars have been published; however, large parts deal with the synthesis and biological properties of multivalent piperidine iminosugars rather than pyrrolidine [12,14,71,72]. Moreover, the latter research field was scarcely reviewed in the decade of its rapid development (2012–2022) [73]. The present review illustrates the detailed synthesis and multivalent effects of all multivalent pyrrolidine iminosugars that have been assayed against various glycosidases and provide an overview of the main achievements made to date.

## 2. Syntheses of Multivalent Pyrrolidine Iminosugars

### 2.1. Synthesis of Di- and Trivalent Iminosugars

The first synthesis of tethered di- and trivalent pyrrolidine iminosugars to interfere with carbohydrate processing enzymes was reported by Robina and co-workers in 2013 [70]. At that time, the advantages of the multivalent effect for glycosidase inhibition over the corresponding monomer were gradually realized (e.g., for α- and β-glucosidases [74], for β-galactosidases [75]). Taking advantage of their experience in designing glycosidase inhibitors [76,77], the authors investigated the multivalent approach by comparing the α-l-fucosidase inhibitory activities of multi- and mono-pyrrolidine iminosugars. For this purpose, four di- and trivalent pyrrolidine derivatives (**1**–**4**) were synthesized based on *fuco*-configured 1,4-imino-cyclitols **5** and **6**, which displayed good inhibitory activity towards α-l-fucosidase (Figure 3). The benzylamino pyrrolidine **5**, designed as a monovalent reference for dimer **1** and trimers **2** and **3**, was synthesized for the first time, while the furyl-substituted pyrrolidine **6**, previously reported by the same research group [78], was selected for comparison with trimer **4**.

The authors employed a classical amide coupling reaction to synthesize the desired iminosugars, starting from commercially available benzylamine and the *O*- and *N*-protected carboxylic acid **7** [70], using PyBOP as the coupling agent and DIPEA as base gave protected intermediates. Then, the excess benzylamine was easily separated by column chromatography. A similar method was used to remove excess amines for the preparations of **1**–**4**. Finally, isopropylidene deprotection with HCl and subsequent catalytic hydrogenation with H_2_/Pd/C gave the corresponding target product **5** in 84% yield. Di- and trivalent iminosugars were obtained with commercially available *m*-xylylenediamine **9** and triamine **10** as the scaffolds for bi- and trivalent glycomimetics, respectively. Moreover, the long-tethered triamine **11** reported by the same research group [79] was chosen as C-3 symmetric template to yield the long spacer trivalent iminosugar **3**. Amide coupling reactions between scaffolds and compound **7** under the same condition above gave the corresponding target products **1**–**3** in moderate-to-good yields. Similarly, trimer **4** was obtained by the coupling reaction between a previously synthesized pyrrolidine-furan carboxylic acid **12 [78]** and template **10**, followed by standard reductive hydrogenation in 46% yield (Figure 1).

To further understand the complicated multivalent effect on α-fucosidase inhibition, Behr, Robina, and co-workers reported a library of divalent pyrrolidine iminosugars **13**–**17** using polyamine and triazole benzene as spacers to evaluate the contributions of the length and rigidity of the bridge, the number of nitrogen atoms present, and the moieties close to the pyrrolidine to the biological activity of divalent inhibitors [80]. Since there is no report of the monovalent references **18**–**20**, their synthesis routes were also introduced. The inhibitory effect of chemically diverse spacers in dimers on α-fucosidase was systematically investigated, and a potent and specific α-fucosidase inhibitor (compound **17**, Ki = 3.7 nM) was thus discovered (Figure 4).

The target monovalent inhibitors **18** and **19** were synthesized from the known allyl-pyrrolidines **21** and **24 [81]**. Starting from the (2*R*)-configured **21**, after dihydroxylation, oxidative cleavage with NaIO_4_ gave a stable intermediate **22**. Pyrrolidinyl ethanol **18a** was obtained in 37% yield by reducing **22** with sodium borohydride, followed by deprotection with hydrogen and acidification in three steps. The congener **18b** was obtained by reacting **23** with benzylamine followed by deprotection with H_2_/Pd–C (10%), in 43% yield. However, the stereoisomers **19a** and **19b** of compounds **18a** and **18b** could not be obtained by the same synthetic route through (2*S*)-configured **24**, mainly due to the key intermediate **25** after reaction with **24**. The authors reported that the aldehyde **25** would go through an epimerization process, which spontaneously opened the pyrrolidine ring to form the conjugated aldehyde **22**, impeding the synthetic purpose [81,82]. Alternatively, protection of the amino group by switching from Bn to Boc solved this problem and gave the clean and stable (2*S*)-configured **26**. Target iminosugars **19a** and **19b** were afforded by reduction and reductive amination under the same conditions as introduced above (Figure 2).

The synthetic routes for dimers **13** and **16** were similar to those of their corresponding monomers. An excess of compound **26** (2.2-fold) reacted with hexamethylene-diamine or spermine to produce the corresponding di-imines, which were then reduced via sodium borohydride. Silica gel chromatography was employed to separate the excess **26** and yield the corresponding protected dimers **13** and **16**, which were further deprotected to give the target products. It is worth noting that under this method, homologue **16** with a spermine bridge contained some impurities. Hence, a further sequence of Boc protection/purification/deprotection (MeOH: HClaq) was required to obtain pure **16** in 29% yield (Figure 2).

The synthesis of the divalent iminosugars **14** and **15** was started from the known compound **28** [83], which was reacted with benzylamine and sodium triacetoxyborohydride, followed by acidification and deprotection to afford compound **15a** in 26% yield. Dimer **15b** was obtained by debenzylation of **15a** under the H_2_/Pd/C system in 65% yield. Reacting **28** (2-fold excess) with hexamethylenediamine by similar methods (MgSO_4_ then NaBH_4_ or amine then NaBH(OAc)_3_) both gave dimer **29** in a low yield (25%) with an undesired trivalent product **31** (27%). Gratifyingly, the yield of **29** could be increased to 49% by reacting **28** with ethylenediamine in the presence of sodium borohydride and 2,2,2-trifluoroethanol. For exploration, the amino-protected dimer **29** and trimer **30** were both deprotected under hydrochloric acid to generate the target products **14** and **31**, which were likewise tested towards α-fucosidase. Dimer **17** was synthesized due to the good inhibitory activities of (pyrrolidin-2-yl)triazoles shown by the researchers previously [83]. Thus, after the reduction of **28** to **32**, target dimer **17** was generated through a two-step reaction by treating excess **32** with 1,3-bis(azidomethyl)benzene under the catalysts of CuI and DIPEA, followed by acidification. Column chromatography was carried out to remove unreacted **32** and yield **17** in 44% yield (Figure 3). Increasing results began to highlight the advantages of multivalent effects. However, some contradictory experimental results were still reported. Elucidating the specific binding mechanisms of multi-ligands with enzymes is urgent and challenging. Behr and co-workers reported three stereoisomeric pyrrolidine dimers in 2016 to explore the divalent effect on fucosidase inhibition [84]. The divalent iminosugars (**33**, ***ent*-33**, and ***meso*-33**) were constructed based on a known fucosidase inhibitor **34** reported by Steensma [85]. The monovalent iminosugars **35**, ***ent*-35**, and **36** were also synthesized as referenced (Figure 5).

Known compound **37** and its enantiomer ***ent*-37** were used as starting materials for synthesizing homodimer **33** and its enantiomer ***ent*-33**, respectively [86]. Hemiacetal ***ent*-37** was converted to **38** by amination with benzylamine. The intermediate **38** was then subjected to a highly stereoselective ethynylmagnesium bromide-mediated nucleophilic addition to aminoalcohol ***ent*-39** in 70% yield for two steps [87]. Then, azide ***ent*-40** was obtained in 68% yield from ***ent*-*39*** upon intramolecular nucleophilic reaction in the presence of MsCl, which was employed to activate the secondary hydroxyl to invert the configuration at C(OH). Homodimerization was carried out simply via the oxidation coupling of pyrrolidine ***ent*-40** using Pd(PPh_3_)_2_Cl_2_ and CuI as catalysts in the presence of i-PrNH_2_ to generate the diyne ***ent*-41** in 84% yield. Finally, the target homodimer ***ent*-33** was prepared in 38% yield by alkyne reduction, hydrogenolysis of the benzyl groups, and acidolysis of ***ent*-41**. Monomer ***ent*-35** was readily prepared by the same reduction/acidolysis sequence from ***ent*-40**. The same synthetic route was applied to the known **37** to prepare compounds **33** and **35** (Figure 4).

The *meso* analog ***meso*-33** cannot be obtained by coupling ***ent*-40** with its enantiomer directly, due to the formation of hard-to-remove mixture ***ent*-41/41**. In order to avoid reaction monitoring and isolation problems, ***ent*-40** and known *N*-allyl protected enantiomer **42** [88] were employed to obtain ***meso*-33** through the same way used to generate **41** described above. An excess of enantiomer **42** was necessary to decrease the production of compounds ***ent*-41** and **43**. As expected, the target hetero-diyne **44** was obtained and isolated in high yield (61%). Cleavage of the *N*-allyl group from **44** in the presence of NDMBA and Pd(PPh_3_)_4_, followed by hydrogenolysis of the benzyl groups and final acidolysis, afforded the target dimer ***meso*-33**. Monomer **36**, an analog of **33** whose second pyrrolidine moiety was replaced by a phenyl group, was prepared from the known diyne **46** [53] using classic hydrogenation (H_2_, Pd/C, MeOH) in 93% yield (Figure 5).

Two years later, Moreno Vargas and co-workers pioneered a valuable methodology for rapid, efficient screening of the divalent inhibitors to α-fucosidases and β-galactosidase, as well as studying the multivalent approach in the inhibition of glycosidases [89]. The Cu(I)-catalyzed alkyne-azide cycloaddition (CuAAC) reaction, a fantastic chemical reaction based on Huisgen 1,3-dipolar cycloaddition chemistry [90,91] and then developed by Meldal [56] and Sharpless [57], was employed to generate three libraries of divalent iminosugars (**47a**–**l**, **48a**–**l**, **and 49a**–**l**) between alkynyl pyrrolidines **47**–**49** and the set of diazides **a**–**i**. Due to the high efficiency of the CuAAC reaction, the obtained crude products could be directly screened for enzyme inhibitors without purification. It is worth noting that the discovery of the CuAAC reaction extensively promoted the development of the multivalent approach (Figure 6).

Alkynyl pyrrolidines **47**–**49** were selected as the skeletons because their analogs (pyrrolidin-2-yl)triazole and (pyrrolidin-2-yl)furans were previously shown to exhibit significant glycosidase inhibition to α-fucosidases and β-galactosidases by the same group [83,92]. The initial step was to prepare the different tethered alkynyl pyrrolidine derivatives **47**–**49**. Known compounds **32 [84]** and **53** [93], previously prepared by the same group from _D_-lyxose and _D_-glucose, were employed as starting materials. Alkynyl pyrrolidine **47** was readily prepared from **32** via Boc-deprotection in TFA in 87% yield. As expected, the CuAAC coupling reaction was carried out between alkynyl pyrrolidine **32** and azide **50** [94] to yield quantitative triazole **51**. Then, propargylation of triazole **51** with NaH followed by acidic deprotection of derivative **52** quantitatively provided the desired alkynyl derivative **48**. Classic amide coupling conditions (PyBOP/propargylamine) were employed to form the epimers **54a** and **54b** from compound **53**, further separated by chromatography. Based on the previous report, *cis*-configured epimer **54b** was chosen for deprotection to yield the target pyrrolidine-furan hybrid **49**, since *trans*-configured epimer **54a** exhibited weak α-fucosidase inhibition. Finally, the diazides **a**–**l**, another part for CuAAC coupling, were prepared according to the design of the spacer. (Pyrrolidin-2-yl)triazole libraries were generated by parallel CuAAC couplings between alkynyl functionalized pyrrolidines **47**–**49** and diazides **a**–**l** under the catalysts of CuI or CuSO_4_. Due to the high efficiency of CuAAC coupling, granting almost quantitative yields with no side reactions, the desired products were all processed and directly screened for enzyme inhibition testing (Figure 6).

Crude screening indicated that dimer **47i** was the best inhibitor of α-fucosidases from the bovine kidney (k_i_ = 0.15 nM), and that dimer **49e** was the best inhibitor of β-galactosidase from the bovine liver (k_i_ = 5.8 µM). Hence, compounds **47i** and **49e** were scaled up for detailed and complete analysis. To evaluate the multivalent effect on enzyme inhibition, monovalent references **56**–**59** were prepared. The synthesis routes adopted the same reaction conditions with compound **51** and CuAAC cycloaddition, and therefore will not be repeated here (Figure 7).

The generation and in situ bio-screening of compound libraries mediated by efficient chemical reactions such as click reactions has proven to be an economical, rapid and efficient screening method for enzyme inhibitors, mainly in the context that the spatial structure of most enzymes is still unknown. More recently, Moreno and co-workers continued their work by screening a library of divalent pyrrolidine iminosugars to find inhibitors of human hexosaminidase [69]. A nanomolar and remarkably selective inhibitor of human nucleocytoplasmic β-*N*-acetylglucosaminidase was thus discovered.

The authors selected pyrrolidine derivative **60** as the skeleton of the divalent iminosugar libraries. Compound **60** was proved to be an outstanding inhibitor of β-*N*-acetylhexosaminidase in 2001 by Wong’s group [95], which is consistent with the purpose of this research. Amino and azido functional groups were introduced based on **60** through molecular modification to carry out the subsequent click reaction. Azide **63** was obtained via reduction of cyanide in the known compound **61 [96]**, followed by acylation and reductive amination with 6-azidohexanal using NaBH_3_CN as catalyst in 49% yield (Figure 8) [97]. Catalytic hydrogenation of **63** by H_2_/Pd/C gave the amine **64** in 76% yield (Figure 8).

A sub-library **I** was generated via CuAAC reaction between **63** (2.4–2.5 equiv.) and dialkynes **a**–**e** (1.0 equiv.) in the presence of CuSO_4_·5H_2_O (0.14 equiv.), sodium ascorbate (0.44 equiv.), and *t*-BuOH/H_2_O with high yield (Figure 9). In parallel, (thio)urea-bond forming reactions between compound **64** (2.4–2.5 equiv.) and diisothiocyanates **A**–**E** (1.0 equiv.) were carried out in solvent DMSO to give a sub-library **II** (Figure 10). Finally, the crude divalent iminosugars **63a**–**e** and **64A**–**E** were assayed as β-*N*-acetylglucosaminidase inhibitors, and thus compounds **63A** and **64D**, with the highest inhibitory potency, were screened and studied in detail. Similar to the previous protocol [89], the inhibition potency of divalent iminosugars was compared with corresponding monomers to evaluate the multivalent effect. Compound **65**, the reference of **63A**, was synthesized by CuAAC cycloaddition between **63** and methyl propargyl ether in 61% yield. Similarly, compound **66**, as control of **64D**, was generated through (thio)urea-bond forming reactions between **64** and phenyl isothiocyanate in 62% yield (Figure 11).

As described above, several examples of multivalent pyrrolidine iminosugars were successfully prepared and used for biological activity exploration through the efficient CuAAC reaction. However, this reaction also brings some problems. For example, the catalyst copper ion required has a high chance of complexing with multiple nitrogen atoms in the triazole produced by the reaction, increasing the risk of metal ion contamination [98]. Moreover, the CuAAC reaction is usually carried out in the last step between the monovalent skeleton and scaffold, which limits the choice of monomer part of the final iminosugars. Therefore, developing new strategies without metal catalysts for the preparation of bio-related iminosugars is highly desirable [61].

Cardona and co-workers reported an alternative way to synthesize multivalent pyrrolidine iminosugars without metals in 2019 [62]. The synthesis relies on iminosugar pyrrolidine DAB and three selective and high-yielding steps (1,3-dipole cycloaddition with nitrone **67**, *N–O* bond cleavage of the adduct, and selective *N*- and/or *O*-allylation), and allows the preparation of different topologies in the DAB clusters. Nitrone **67**, obtained from the commercially available tribenzylated d-arabinose [99], and allyl benzyl ether **68 [100]** were employed to prepare *exo-anti* isoxazolidine **69** in 85% yield through 1,3-dipole cycloaddition (1,3 DC). The 1,3 DC process is a crucial step since a high degree of stereoselectivity in the reaction must be guaranteed to reduce isomer formation. Previous research showed that high *exo-anti* selectivity is ascribed to the *tans-trans* configurated nitrone **67**, whose C-3 and C-5 substituents on the same face are opposite to C-4. Thus, the *exo* mode was preferred to avoid repulsive steric interactions with a substituent at C-4 [99,101,102]. Then, the cleavage of the *N–O* bond of **69** in the presence of 10 equiv. of Zn afforded **70** quantitatively. Note that compound **70** is a key intermediate since the selective *N*-and/or *O*-allylation would generate new dipolarophiles **71**–**73**, which would introduce a second or third DAB moiety by 1,3 DC with nitrone **67**. MW-assisted selective *N*-allylation of **70** was carried out using electrophile allyl bromide to afford intermediate **71** in 86% yield. Protecting the amine of **70** using benzyl bromide, followed by selective *O*-allylation in the presence of allyl bromide and K_2_CO_3_, gave intermediate **72** in 71% yield. The *N*,*O*-bis allylated pyrrolidine **73** was obtained by treating **70** with a high excess of allyl bromide (6 equiv.) and NaH (8 equiv.) in 81% yield (Figure 12).

The synthesis of multivalent DAB iminosugars was carried out as initially designed. 1,3 DC reaction between **71** and nitrone **67**, followed by catalytic hydrogenation with Pd/C in MeOH/HCl, gave the bis-pyrrolidinium hydrochloride **75** in quantitative yield, which was submitted to the ion exchange resin Dowex 50WX8–200, followed by treatment with the strongly basic Ambersep 900-OH resin to afford the divalent DAB-based iminosugar **76** in 40% yield (Figure 13). A similar approach was applied to the preparation of dimer **79** and trimer **82** from corresponding intermediates *O*-allylated **72** and *N*,*O*-allylated **73** (Figure 14). After three steps—1,3 DC, catalytic hydrogenation, and ion exchange resin—DAB-based iminosugars **79** and **82** were generated in 95% and 60% yield.

### 2.2. Synthesis of Multivalent Iminosugars

In 2016, Cardona and co-workers explored the inhibition of sulfatases using the first two examples of pyrrolidine clusters [67]. Nonavalent pyrrolidine iminosugars **83** and **84** were obtained from DAB and 1,4-dideoxy-1,4-imino-d-ribitol (**86**), prepared by the deprotection of the starting nitrones **67 [103]** and **85 [104]**. Upon selective *N*-alkylation with 1-azido-6-bromohexane [105] in the presence of K_2_CO_3_ under MW irradiation, DAB gave the deprotected azide **87** in 92% yield. Reacting **87** with the nonadiyne scaffold **89** [106] using the standard CuAAC cycloaddition condition (CuSO_4_ (30 mol%), sodium ascorbate (60 mol%), THF/H_2_O) afforded a mixture of unreacted **87** and desired product **83**, which was further purified by column chromatography and size exclusion chromatography Sephadex LH-20, obtaining pure **83** in 81% yield. Similar synthetic routes were applied for the synthesis of target iminosugar **84**. Starting from a different configuration of bioactive 1,4-dideoxy-1,4-imino-d-ribitol (**86**), selective *N*-alkylation afforded **88**, and CuAAC coupling gave ribose configured **84** in 71% yield (Figure 15).

The monovalent references **90** and **91** corresponding to compounds **83** and **84** were synthesized to explore the multivalent effect further (Figure 16). The reference **90** was synthesized from known d-arabinose derived nitrone **67**. Two-step reduction by NaBH_4_ followed by Zn in AcOH gave amine **92** in 98% yield, which was treated with 1-azido-6-bromohexane in basic conditions under microwave irradiation to afford azide **93** in 88% yield. The CuAAC coupling between azide **93** and 3-butyn-1-ol was carried out under MW irradiation to give **94** (93%), which was further deprotected using H_2_ and Pd/C in acidic MeOH to give the final product **90** in 74% yield. Compound **91**, generated through one-step synthesis, was obtained by cycloaddition of azide **88** to 3-butyn-1-ol in the presence of CuSO_4_ and sodium ascorbate under MW irradiation in 89% yield (Figure 16).

Results showed that the nonavalent pyrrolidine iminosugar **83** exhibited impressive inhibition of *N*-acetylgalactosamine-6-sulfatase (GALNS). Considering that the DAB motif in **83** is a widely available glycosidase inhibitor, and the fact that GALNS and α-mannosidases both have dimer properties, the same research group continued to explore the interaction of α-mannosidases with different multivalent architectures based on iminosugar DAB in 2017 [107].

Similar to the method for synthesizing compound **83**, the CuAAC cycloaddition of intermediate azide **87** and multivalent alkyne scaffolds was exploited to generate new tetra- and trivalent pyrrolidine iminosugars **96** and **98** (Figure 17). Microwave-assisted CuAAC of azide **87** (3.5 or 4.5 equiv.) with trivalent scaffold tris[(propargyloxy)methyl]amino-methane **97** and tetravalent scaffold **95** (1.0 equiv.) in the presence of CuSO_4_ (0.3 equiv.) and sodium ascorbate (0.6 equiv.) in THF/H_2_O a, followed by purification through flash column chromatography and size-exclusion chromatography Sephadex LH-20 (H_2_O) to separate the excess azide **87**, gave the pyrrolidine iminosugar clusters **98** (48%) and **96** (76%) in good yields. In addition, inhibitory performance against a panel of glycosidases was evaluated among the nona-, tetra-, and trivalent iminosugars **83**, **96**, and **98**, as well as monovalent references DAB and its derivative **90**.

Two years later, Moreno Vargas and co-workers prepared four multivalent pyrrolidine iminosugars for GH1 β-glucosidases A and B (BglA and BglB) to continue their study on binding modes and key determinants responsible for the inhibitory effect displayed by pyrrolidine-based clusters [108]. Therefore, as before, the CuAAC click reaction between pyrrolidine-azide derivatives (**99** and **101**) and two different tri- or hexavalent alkynyl spacers (**97** and **102**) was exploited to give the target clusters (**103**–**106**). It was found that spacers containing aromatic moieties in multivalent inhibitors showed excellent inhibition against octameric BglA (µM range) compared to the similar monomeric BglB. Moreover, a modest multivalent effect was detected for the hexavalent inhibitor **106**.

Starting from the azidomethyl pyrrolidine **107** reported by the same group, the protected azide **99** was obtained in good yield by reacting with the aromatic alkyne **108** [109] through CuAAC and then treating with NaN_3_ in 93% yield. Then, classic conditions (HCl/THF) were used to get rid of the protecting group to give the unprotected derivative **100** quantitatively. The same reactions performed in **107** and alkyne **109** [95] via a sequence of CuAAC coupling, nucleophilic displacement with NaN_3_, and acidic deprotection afforded **101** in high yield. The synthesis of azides **100** and **101** was not only to obtain the final multivalent clusters but also as monomers for bioactivity control. However, due to the poor solubility of azide **100**, protected azide **99** was selected in the subsequent synthesis and then deprotected again with hydrochloric acid (Figure 18).

Finally, microwave-assisted CuAAC cycloaddition of azide **99** or **101** with scaffolds **97** [110] and **102** [108] in the presence of CuSO_4_, sodium ascorbate in THF/H_2_O (2:1), followed by checking reactions through ^1^H NMR spectra of the crude mixtures, gave the pyrrolidine-based iminosugar clusters **103**–**106** in good yields (57–93%, Figure 19). It is worth noting that the resulting crudes, mixed with excess azide **99** or **101** and desired products, were purified by stirring with Quadrasil^®^ MP followed by chromatography column (silica gel or Sephadex LH-20).

Moreno Vargas and co-workers have long worked on the design and synthesis of mono- and multivalent iminosugars to evaluate their inhibition activities toward various disease-related enzymes. Because two enzymes, β-glucocerebrosidase (GCase) and α-galactosidase (α-Gal A), are involved in Fabry and Gaucher diseases, respectively, and combined with the experimental results that pyrrolidine-3,4-diol skeleton-based iminosugars exhibit bioactivity to human lysosomal GCase reported by his group [111,112], exploring the multivalent effect on these two enzymes became their target. The author reported four sets of multivalent pyrrolidine iminosugars with different valency, configuration, and spacers to perform a systematic analysis of the inhibition of the lysosomal glycosidases in 2020 [49].

The six azidoalkyl pyrrolidines shown in Figure 7 were selected as monovalent references as well as anchoring moieties for CuAAC coupling reactions. Compounds **100**, **101**, **110**, and **111** were all known compounds reported by the same group [108,111], while **112** and **113** were newly synthesized. The synthetic routes of **112** and **113** were the same as the preparation for their epimers **101** and **100** described in Figure 18 [108]. The known scaffolds shown in Figure 8 were selected to synthesize the tri-, tetra-, hexa- and nonavalent iminosugars via CuAAC coupling with azido derivatives.

General reaction conditions of the click reaction involved using CuSO_4_ and sodium ascorbate as catalysts and in THF–H_2_O under microwave irradiation at 80 °C. Similar to the methods for compounds **103**–**106**, chromatography column (silica gel or Sephadex LH-20) was employed to separate the unreacted azidoalkyl pyrrolidines. Multimeric derivatives (**114**–**133**) were generated in high yields (55–99%). The monovalent references (**134**–**139**) were prepared via CuAAC reaction between monomers (**110** and **111**) and the corresponding alkynes in high yields (79–91%), which were submitted to evaluate inhibition against human GCase and α-Gal A along with the corresponding multivalent iminosugars **114**–**133** (Figure 9, Figure 20).

More recently, Gaeta, Cardona, and co-workers constructed pyrrolidine-based multivalent clusters, employing the less researched scaffold resorcinarene [113], which exhibited conformationally mobile ability [114,115,116], to explore the role of both conformability and valency in the inhibition of therapeutically relevant enzyme Golgi α-mannosidase IIb (GMIIb) [68].

Resorcinarene **140**–**142** [113,116] are macrocycles consisting of four to six rings of resorcinol obtained by resorcinol/aldehyde acid-catalyzed condensation reaction (Figure 21). Interestingly, each aromatic moiety of the macrocycle contains two hydroxyl functional groups, making it suitable for constructing multivalent iminosugars. Azides **87** and **143**, belonging to the DAB-derived pyrrolidine family, were selected as skeletons (Figure 21). Azide **87 [67]** was reported by the same group previously, and new azido-ending ligand **143**, which possessed a more hydrophilic linker, was newly synthesized to explore the role of the nature of linkers in bioactivity [68].

Different scaffolds *C*-methyl-resorcin [4] arene **140**, resorcin [4] arenes **141**, and resorcin [6] arenes **142** were allowed to react with propargyl bromide 2 equivalents per hydroxyl in acetone by treatment with excess K_2_CO_3_ to give alkyne-ending scaffolds **144**–**146** in high yields (58–98%). Then, target resorcinarene-based iminosugars **147**–**149** were obtained by reacting azide **87** with scaffolds **144**–**146** through the CuAAC click reaction in moderate yields (27–44%). The unsubstituted azides **87** and **143** were purified through chromatographic column (silica gel, gradient: from MeOH to ammonia solution 4 M in MeOH). Finally, the compound **150**, featuring a more hydrophilic linker, was obtained by CuAAC reaction between azide **143** and scaffold **146** (Figure 21). It is worth pointing out that, although the valency of **147** and **148** was the same, the scaffold resorcin [4] arene (**140**) was conformationally blocked in a cone conformation, thanks to the presence of CH_3_CH bridges between aromatic rings [117]. As a result, iminosugar **147** was more flexible than **148**.

## 3. Biological Activity of Multivalent Pyrrolidine Iminosugars

### 3.1. Inhibition of α-Fucosidases

α-Fucosidases (AFU) are lysosomal acid hydrolase enzymes that catalyze the hydrolysis of α-fucose units located on the cell surface oligosaccharides and participate in various biological processes, including immune response, signal transduction, and antigenic determination [118,119,120]. Changes in the activity of AFU in serum or tissue significantly correlate with the occurrence of tumors, such as hepatocellular carcinoma [121], colon adenocarcinoma [122,123]_,_ and gastric cancer [124]. Since the discovery of pyrrolidine 1,4-iminocyclitols as potent inhibitors of AFU [77,78], Robina and co-workers first explored the AFU-inhibitory activity of di- (**1**) and trivalent pyrrolidine iminosugars **2**–**4** in 2013 (Figure 3) [70]. Results showed that all the newly synthesized compounds displayed high AFU inhibition (IC_50_: 1.6–17 µM) and excellent selectivity. However, compared with the monomer references **5** and **6** (Figure 3), the effect of multivalency was not convincing, except for the trivalent iminosugar **2** (K_i_ = 0.3 µM, Table 1), which showed seven-fold more potent inhibition activity than monovalent reference **5** (K_i_ = 2.1 µM). Compounds **2** and **3** (K_i_ = 0.4 µM, Table 1) displayed almost equivalent activities, indicating that the increase in the length of multivalent iminosugars was not clearly linked to the inhibitory properties of the enzyme. Divalent iminosugars with more diverse spacers were also reported subsequently [80]. Polyamino and triazole-benzyl bridged iminosugars (**13**–**17**, Figure 4) were constructed to develop potential inhibitors for AFU. Dimers **13**, **14**, and **16** showed stronger inhibition than their corresponding monomers, while compounds **14**, **15**, and **31** yielded the opposite results. Triazole-benzyl bridged iminosugar **17** showed excellent enzyme inhibition to AFU (IC_50_ = 74 nM, K_i_ = 3.7 nM, Table 1) while dimer **13** (IC_50_ = 1.2 μM, Table 1) indicated the existence of multivalency compared with its control **19b** (IC_50_ = 13 μM, Figure 4), a 10.8-fold potency enhancement. The result that compound **17** exhibited excellent inhibition toward AFU was consistent with the fact that the presence of an additional aromatic or heteroaromatic binding component close to the five membered iminocyclitols notably increases their inhibitory activity to AFU, which was shown by Robina [83], Behr [125], and Wong [126]. To further explore the ligand-enzyme binding modes, stereoisomeric pyrrolidine dimers (**33**, ***ent*-33**, and ***meso*-33**, Figure 5) with short and flexible space were synthesized [84]. Dimer **33** showed potential inhibition of AFU (IC_50_ = 0.108 µM, K_i_ = 23 nM, Table 1) when compared to its monovalent reference **35** (IC_50_ = 2.0 µM, K_i_ = 0.18 µM, Figure 5), which to some extent confirmed the existence of the multivalent effect. The divalent ***meso*-33** also showed potential inhibition of AFU (IC_50_ = 0.365 µM, K_i_ = 0.051 µM), while compound ***ent*-33** was significantly less potent (IC_50_ = 84 µM, K_i_ = 12 µM). Through detailed controlled trials and structural analysis, the authors suggested that the inhibition enhancement obtained with divalent compounds could be explained by additional interactions of the hydrophobic moiety with a lipophilic binding pocket other than the active site. This hypothesis was confirmed by the 3-D structure of the bacterial fucosidase *Bt*Fuc2970 complexed with the best divalent inhibitor **33**. However, other mechanisms such as rebinding could not be completely ruled out. In 2018, Moreno Vargas and co-workers successfully screened a batch of AFU inhibitors through the CuAAC click reaction followed by in situ biological screening and identified one of the most effective enzyme inhibitors, **47i** (IC_50_ = 48 nM, K_i_ = 15 nM, Figure 6, Table 1) [89]. The higher inhibition shown by dimer **47i** compared to its analogue **48i** could be argued to be due to non-specific interactions of the diphenylsulfone spacer in the loop regions near the GH29 family’s enzymatic active site. Due to controversy over the reference selection, a valid multivalent effect could not be given, but the discovery of compound **47i** proved the rapidity and efficiency of the methodology, which should be highlighted in the screening of enzyme inhibitors. To some extent, these results indicated that the multivalent effect of iminosugars on α-fucosidases is probably due to the additional unspecific interactions with a noncatalytic subsite, which would be beneficial to medicinal chemists in the rational design of α-fucosidase inhibitors.

### 3.2. Inhibition of α-Mannosidases

α-Mannosidases are mainly involved in the biosynthesis and catabolism of *N*-glycans in cells. Such processes are, for instance, involved in the treatment of cancers and lysosomal diseases [127,128,129]. The first evidence of the multivalent effect on iminosugars was gained through the interaction between Jack bean α-mannosidase (JBMan) and a trivalent DNJ conjugate [58]. Due to the successful analysis of its crystal structure and the ease of purchase, JBMan has become the most investigated enzyme for multimeric inhibition studies [48]. Novel tri-, tetra-, and nonavalent pyrrolidine iminosugars (**98**, **96**, and **83**, Figure 15 and Figure 17) were constructed by Cardona and co-workers to investigate the binding modes to α-mannosidases [107]. A large multivalent effect was observed from the three iminosugars (rp/n >> 1). The DAB-based nonavalent iminosugar **83** (Figure 15) was the best inhibitor of JBMan (IC_50_ = 95 nM, Table 1), with a 13,684-fold (rp/n = 1520) stronger inhibitory potency than the corresponding the monovalent reference **90** (IC_50_ = 1300 µM, Figure 16). The trivalent compound **98** and the tetravalent **96** also showed good multivalent effects towards JBMan, with rp/n values of 46 and 10, respectively. Transmission electron microscope (TEM) analysis, nuclear magnetic resonance (NMR), and molecular dynamic studies were carried out to elucidate the binding mode of the multivalent iminosugars and α-mannosidases. NMR studies showed the existence of specific interactions of the multivalent ligands with JBMan, which presumably take place within the enzyme active site. TEM studies indicated that the binding mode would probably be intermolecular cross-linking, due to the formation of ligand–JBMan aggregates. It is worth noting that a remarkable selectivity of iminosugars (**83**, **96**, and **98**) for Golgi α-mannosidase IIb (GMIIb) over lysosomal α-mannosidase II (LManII), two biologically relevant enzymes (GMIIb: tumor growth and cell metastasis; LManII: disorder mannosidosis), was observed. The interesting selectivity appeared particularly relevant for selective application of multivalent compounds in anticancer therapy without the undesirable side effect of mannosidosis syndrome. Subsequently, scaffold resorcinarene was employed to explore the role of both the conformability and the valency of multivalent iminosugars to therapeutically relevant target GMIIb [68]. Similarly, both the 8-valent (**147**, **148**, Figure 21) and 12-valent iminosugars (**149**, **150**, Figure 21) exhibited greater selectivity to JBMan and GMIIb over LManII. Biological assay indicated that 12-valent **149** had stronger inhibition, for example, towards GMIIb (IC_50_ = 0.7 µM, Table 1) than 8-valent **147** (IC_50_ = 3.7 µM, Table 1) and **148** (IC_50_ = 5.3 µM, Table 1), which further showed that the inhibitory activity of resorcinarene-based conjugates was related to their valency. The 12-valent iminosugar **150**, possessing a more hydrophilic group, showed weaker inhibition (GMIIb, IC_50_ = 28.5 µM, Table 1) than the same valent **149** (GMIIb, IC_50_ = 0.7 µM, Table 1). This was ascribed to the unfavorable repulsions between oxygen atoms on the linker with electron-rich atoms of the amino acid residues of the GMIIb protein. In addition, the 12-valent **149** showed a remarkable multivalent effect towards JBMan (IC_50_ = 1.2 µM) compared to its monovalent reference **90** (Figure 16, IC_50_ = 1300 µM, rp/n = 90). Computational studies suggest that the binding mode should be the rebinding process, since the resorcinarene ligands bind the dimer of the JBMan by coordination of one Zn ion at a time. From these results, we can know that the multivalent effect of pyrrolidine iminosugars on α-mannosidases has a great relationship with valency. Generally, higher valency iminosugars usually exhibit better inhibitory activities. The multivalent effect is also affected by the type of linker and the conformation of the scaffold. In addition, the proposed binding modes—cross-linking and aggregation, and bind and recapture (Figure 2)—are more likely involved in better responses of multivalent pyrrolidine iminosugars toward α-mannosidases. However, a binding mode that involves both the active site and non-catalytic subsites cannot be completely excluded.

### 3.3. Inhibition of Other Disease-Related Glycosidases

Besides the α-fucosidases and α-mannosidases introduced above, Cardona and co-workers explored the impact of multivalency on sulfatases involved in lysosomal storage disorders (LSD) for the first time [67]. A decrease in two lysosomal enzymes, *N*-acetylgalactosamine-6-sulfatase (GALNS) and iduronate-2-sulfatase (IDS) could cause diseases of mucopolysaccharidoses: Morquio A syndrome and Hunter disease, respectively [130,131,132]. Nonavalent DAB-based iminosugars **83** (GALNS: IC_50_ = 47 µM, IDS: IC_50_ = 140 µM, Figure 15, Table 1) and **84** (GALNS: IC_50_ = 85 µM, IDS: IC_50_ = 31 µM, Figure 15, Table 1) exhibited strong inhibition to both enzymes compared to the negligible monovalent references **90** (GALNS: IC_50_ = 3900 µM, IDS: IC_50_ = 3200 µM, Figure 16) and **91** (GALNS: IC_50_ = 5000 µM, IDS: IC_50_ = 5500 µM, Figure 16). The results demonstrated that a good multivalent effect was achieved with pyrrolidine-based clusters towards sulfatases. For example, **84** showed a remarkable multivalent effect toward IDS (rp/n = 19.7, Table 1). However, detailed kinetic studies and proposed binding modes were not given. On the basis of this result, DAB-based iminosugars **79** (Figure 14) and **82** (Figure 14) with different ligand topologies were synthesized for GALNS inhibition two years later by changing CuAAC coupling to a new strategy which avoided contamination with copper ions [62]. Dimer **79** and trimer **82** showed IC_50_ to GALNS in the low micromolar range (0.3 and 0.2 μM, respectively), confirming that multimerization of DAB epitopes generates potent GALNS inhibitors. Comparing the inhibitory activities of **79** and **82** with **83** and **84**, we can learn that the ligand topology strongly affected the affinity of the DAB-based multivalent iminosugars for GALNS. Divalent iminosugar **64D** (Figure 10) was discovered to be a potent inhibitor of human hexosaminidases, the potential pharmacological targets for drug development, via the screening of two libraries of divalent pyrrolidine iminosugars [69]. The results showed that compound **64D** exhibited remarkable inhibition of human β-*N*-acetylglucosaminidase (hOGA) in the nanomolar range (K_i_ = 6.1 nM, Table 1) compared to the monovalent reference **66** (K_i_ = 47.6 nM, Figure 11). No significant multivalent effect was observed in the inhibition of any of the hexosaminidases by dimers **63** and **64**. However, compound **64D** displayed excellent selectivity towards hOGA compared with human lysosomal β-*N*-acetylhexosaminidases (hHexB, K_i_ = 168 μM, Table 1), with an approximately 27500-fold enzyme affinity enhancement. It was observed very clearly from the result that multivalency could also be a promising tool to modulate the inhibition selectivity of multivalent iminosugars. Similarly, the (2*R*)-nonavalent iminosugar **133** (Figure 9) was screened by Moreno-Vargas and co-workers for human α-galactosidase A (α-Gal A), which is involved a common lysosomal storage disorder, Fabry disease [49]. Compound **133** displayed remarkably potent inhibition and multivalent effect (K_i_ = 0.2 μM, rp/n = 42, Table 1) towards α-Gal A, being a 375-fold more potent inhibitor than the monovalent reference **139** (K_i_ = 75 μM, Figure 20). The author suggested that the multivalent effect was probably due to the involvement of interaction mechanisms such as statistical rebinding, additional binding with allosteric sites, and/or aggregative processes. More importantly, the activity enhancement effect of compound **133** towards α-Gal A in Fabry fibroblasts constitutes the first evidence of the potential of multivalent iminosugars to act as pharmacological chaperones in the treatment of this LSD.

## 4. Conclusions

The last decade has witnessed the rapid development of multivalent effects in glycosidase inhibition and drug discovery. In this review, we systematically summarized the process of fabricating multivalent iminosugars based on pyrrolidine in terms of design strategies, synthesis routes, and glycosidase inhibition investigations. Up to 12-valent pyrrolidine iminosugars were synthesized through classic click reactions, and thus several outstanding inhibitors were discovered. For example, nonavalent inhibitors based on DAB and one of its epimers demonstrated the existence of the multivalent effect in sulfatases for the first time [67]. Moreover, nonavalent iminosugars based on pyrrolidine-triazole moieties exhibited a remarkable multivalent effect on one important therapeutic enzyme, human α-galactosidase A, and constitute the first evidence of a multivalent enzyme activity enhancer for Fabry disease [49]. Despite advances in the design and investigation of multivalent iminosugars based on pyrrolidines, some problems and challenges remain.

The enzymes used for studying the multivalent approach are mostly limited to the more researched models, such as α-mannosidase and α-fucosidase, which means the importance of some therapeutically relevant glycosidases is overlooked. The complex and confusing enzyme–ligand binding mechanism is not a negligible issue when developing new relevant multivalent inhibitors. Elucidating the binding mode(s) would improve glycosidase inhibition efficiency and selectivity, two major problems currently existing. Despite the many challenges, we hope that, with the information presented in this review, researchers in this field will continue to explore multivalent effects based on pyrrolidine for developing new glycosidase inhibitors, as well as for candidates for advanced clinical trials or markets.

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
