# Peer review of "Multivalent Pyrrolidine Iminosugars: Synthesis and Biological Relevance"

_molecules, 2022, doi:10.3390/molecules27175420_

Round 1

Reviewer 1 Report

Please see the attached file entitled “Comments on molecules-1853434“

Author Response

Dear the Reviewer:

Thank you for the rapid handling of this manuscript. We appreciate the Reviewer’ comments and efforts very much. We have tried our best to improve and made some changes in the manuscript as detailed below. The changes in the manuscript have been highlighted to facilitate re-review. We have a point-by-piont response in the attachment.

We hope that you and the reviewer will find our revisions satisfactory.

Best Regards,

Chunyan Liu

College of Pharmacy

North China University of Science and Technology
TangShan 063000 China

chunyanliu@ncst.edu.cn

Reviewer 2 Report

The review provides an overview on select examples of pyrrolidine iminosugars attached to various bi- up to multivalent scaffolds as potential inhibitors of glycosidases (and glycosyl transferases). The concept of multivalency has been successfully applied to lectin-carbohydrate interactions where multiple copies of sugar binding sites are presented in sterically defined arrangements. The molecular details of this effect applied for glycosidases and even for glycosyl transferases is less convincing and the presented data also point towards more or less serendipitous findings from a handful of laboratories in this field. As an example, it is difficult to conceive how multivalency should influence the mechanism of a glycosyl transferase where both sugar nucleotide and acceptor substrates are often bound in a “closed” conformation. The improved inhibitory effects might thus be due to additional but non-specific binding to the surface of the protein, which should have been discussed in more depth in the manuscript.

The chemistry described is well established and follows conventional strategies both for the applied multivalent scaffolds, coupling reactions using click-chemistry as well as modifications and protecting group chemistry of the iminosugars. The authors should have commented on the completeness of the multivalent reactions including separation of undersubstituted derivatives. The methodology used for measuring inhibition constants has not been included (ITC, SPR etc?).

Table 1 should include a column with the literature references

Specific comments:

The Lock-and-Key model referred to (lines 62-69) is outdated as a continuum from rigid binding to induced fit and conformational selection by the protein receptor has been found.

Lines 75&76: The meaning of this sentence is not clear. What is meant by the structural differences between glycosidase and itself (?)?

Line 121: … the carboxylic acid should correspond to compound # 7

Line 131: Benzylamine 8 does not act as a template when forming the amide

Scheme 2: The 100% yield for the 2-step conversion of 26 to 19a is not realistic

Line 164: ..the di-imide should be correctly named as di-imine

Line 228-229: The alkyne-azide cycloaddition was discovered by Huisgen and should be correctly cited.

Line 253: replace…cys by cis

Line 340: % is missing

Scheme 14: the formula for cpd 81 shows number 20?

Line 383: Replace ….truth by …fact

Line 397: Replace …nova by ..nona

Line 479; Replace …valent by valency (same for lines 135, 573)

Line 509: Replace …bonding by binding

Line 533: Typo for iminosugars

Line 550: delete valent

Several additional typos and grammatical errors need to be corrected

References: Absolute configurations of sugars should be indicated with small capital letters (refs. 5, 26, 59, 61)

Author Response

(The authors gave the same response as above.)

Reviewer 3 Report

In this review, the authors have thoroughly discussed about the synthesis and biological relevance of multivalent pyrollidine iminosugars and almost covered the recent developments as well as the historical aspects of the same. This is a much-needed review in this topic.

Comments are presented below.

1. The abstract is presented as per the journal standard, and it completely gives an idea about the manuscript. (No ‘typos’ observed)

2. The authors claim that this is the first of its kind review article thoroughly discussing about the multivalent pyrrolidine iminosugars which is quite acceptable. Ample number of examples are presented.

3. In the introduction section: -

  a) The authors should highlight the importance of multivalent pyrrolidine iminosugars in a separate paragraph.

  b) The pictorial representation of multivalent effect may be improved for better understanding (Fig: 2)

4. In section 2.1 the authors discussed the synthesis of di- and trivalent iminosugars in a systematic manner. Some comments are as follows: -

a) The reaction schemes were drawn correctly with proper reagents and conditions.

b) In the work by Behr, Robina, and co-workers the compound 17 shows the potent and specific α-fucosidase inhibitor property. Is there any specific cause behind this?

c) The cause of epimerization may be presented with proper mechanism (conversion of 24 to 25).

d) Scheme 4; correct the term ‘4 A’; Scheme 5 correct Pb (PPh3)3Cl.

e)  Is there any fucosidase inhibitory activities of 35, ent-35?

5. In section 2.2 the authors discussed the synthesis of multivalent iminosugars in a detailed manner. Some comments are as follows: -

 a) The reaction schemes were drawn correctly with proper reagents and conditions. Most of the examples were based on Click reactions due to its facile nature.

b) Scheme 21 is not clearly understandable.

6. In section 2.3 the authors discussed the biological activity of multivalent pyrrolidine iminosugars in a detailed manner. Some comments are as follows:

a) From the Inhibition of α-fucosidase data, we can we predict the potency of iminosugars and set a trend based on valency? Same observation for α-mannosidase also.

b) The authors may include the proposed mode of action for all types of enzyme inhibition by the iminosugar derivatives.

7. The conclusion section is also presented precisely and covered all aspect of the manuscript from synthesis to biological evaluation of hypervalent pyrrolidine based iminosugars and the future scopes. (no ‘typos’ found)

8.  The authors are recommended to emphasis the importance of iminosugars as an anti-diabetic agent and it is recommended to cite following relevant articles related to iminosugars in introduction section.

  1. Nash, R. J.; Kato, A.; Yu, C-. Y.; Fleet, G. W. J. Iminosugars as therapeutic agents: recent advances and promising trends. Future Med. Chem. 2011, 3, 1513−1521.

  1. Chennaiah, A.; Dahiya, A.; Dubbu, S.; Vankar, Y. D. A Stereoselective Synthesis of an Imino Glycal: Application in the Synthesis of (−)-1-Epi -Adenophorine and a Homoiminosugar. Eur. J. Org. Chem. 2018, 6574−6581.

  1. Chennaiah, A.; Bhowmick, S.; Vankar, Y. D. Conversion of glycals into vicinal-1,2-diazides and 1,2-(or 2,1)-azidoacetates using hypervalent iodine reagents and Me3SiN3. Application in the synthesis of N-glycopeptides, pseudo-trisaccharides and an iminosugar. RSC Adv. 2017, 7, 41755−41762.

  1. Verma, A. K.; Dubbu, S. Chennaiah, A.; Vankar Y. D. Synthesis of di- and trihydroxy proline derivatives from D-glycals: Application in the synthesis of polysubstituted pyrrolizidines and bioactive 1C-aryl/alkyl pyrrolidines. Carbohydr. Res. 2019, 475, 48–55.

  1. Horne, G.; Wilson, F. X.; Tinsley, J.; Williams, D. H.; Storer, R. Iminosugars past, present and future: medicines for tomorrow. Drug Dis. Today 2011, 16, 107−118

Overall, after addressing the points mentioned above, I recommend this review to publish in molecules.

Author Response

(The authors gave the same response as above.)

Reviewer 4 Report

I found this review interesting and informative. The authors summarise the literature very well. The main problem is that the use of English needs improving to make the reading better. I have made marks and comments on the attached pdf. 

Author Response

(The authors gave the same response as above.)

Round 2

Reviewer 2 Report

The authors have done a good job to improve the manuscript. Indeed, the added detailed specific information and discussion on the bioactivities of several multivalent derivatives will allow for a better assessment of the reported results by the readership of the journal and sharpens the context of the review. A final check for a few remaining typos should be done.